# Calculation of soil water content using dielectric permittivity-based sensors; benefits of soil-specific calibration.

Bartosz M. Zawilski, Franck Granouillac, Nicole Claverie, Baptiste Lemaire, Aurore Brut, Tiphaine Tallec

CESBIO Université de Toulouse, CNES, CNRS, INRA, IRD, UPS, Toulouse, 31000 France

*Correspondence to*: Bartosz M. Zawilski (bartosz.zawilski@univ-tlse3.fr)

**Abstract.** Soil Water Content (SWC) sensors are widely used for scientific studies or for the management of agricultural practices. The most common sensing techniques provide an estimate of volumetric soil water content based on sensing of dielectric permittivity. These techniques include: Frequency Domain Reflectometry (FDR), Time Domain Reflectometry (TDR), capacitance, and even remote sensing techniques such as Ground-Penetrating Radar (GPR) and microwave-based techniques. Here we will focus on Frequency Domain Reflectometry (FDR) sensors and more specifically on the questioning of their factory calibration, which does not take into account soil specific features and therefore possibly leads to inconsistent SWC estimates. We conducted the present study in the south west of France, on two plots that are part of the ICOS ERIC network (Integrated Carbon Observation System, European Research and Infrastructure Consortium), FR-Lam and FR-Aur. We propose a simple protocol for soil-specific calibration, particularly suitable for clayey soil, to improve the accuracy of SWC determination when using commercial FDR sensors. We compared the sensing accuracy after soil-specific calibration versus factory calibration. Our results stress the necessity of performing a thorough soil-specific calibration for very clayey soils. Hence, locally, we found that factory calibration results in a strong overestimation of the actual soil water content. Indeed, we report relative errors as large as +115% with a factory-calibrated sensor based on the real part of dielectric permittivity, and up to +245% with a factory-calibrated sensor based on the modulus of dielectric permittivity.

## Introduction

In the context of global warming and the disappearance of water resources, the volumetric soil water content (SWC, also denoted by $\theta$) is one of the most monitored climatic variables as it is a critical interface between all major flows in the water cycle (Wang-Erlandsson et al. 2022). SWC, along with other physical and textural properties of the soil, is key for the estimation of soil water availability and for the study of related processes.

Several techniques have been developed for SWC determination based on direct gravimetric soil sample measurement or indirect measurements. A review of all these techniques can be found in Bittelli, 2011. Most SWC sensors rely on soil dielectric permittivity sensing, because dry soil's relative dielectric permittivity is much smaller than that of pure water (mean values of 4 versus 80; Behari, 2005, Malmberg and Maryott, 1956). From remote-sensing by ground-penetrating radar (Davis and Annan, 1989) or microwave-based measurements (Hoekstra, A. Delaney 1974) to soil sensors, all are based on relative dielectric permittivity determinations. For example, Frequency Domain Reflectometry (FDR) and Time Domain Reflectometry (TDR) SWC sensors rely on a measured signal (frequency or time) that can be related to the dielectric permittivity $\varepsilon$, and hence to the soil water content. These sensors are very widely used and, according to the manufacturers, their accuracy is about 0.03 ($m^3 m^{-3}$) *provided a soil-specific calibration is performed before use*. Sensed relative dielectric permittivity allows to estimate the soil water content, but soil texture and several other soil features should be taken into account for sensor calibration. A brief overview of the used terminology is provided in section 2.

Indeed, as the sample volume is less than 50 $cm^3$, soil heterogeneity may compromise the measurements' reliability. Crack formation, which is common in vertisol, leads to inconsistent measurements. Also, any pebble, vegetable, or animal between the sensor rods affects measurements. Furthermore, like every alternative current (AC)-derived quantity, dielectric permittivity is a complex value, comprising a real part $\varepsilon_R$, and an imaginary part $\varepsilon_I$ (Grimnes and Martinsen, 2015). Ions within the soil greatly affect the dielectric permittivity imaginary part (Campbell, 1990; Szypłowska et al., 2018), especially in the low-frequency range (Skierucha and Wilczek, 2010), that is why the determination of SWC using a sensor based on the dielectric permittivity modulus $|\varepsilon|$ may be less reliable

(Sreenivas et al., 1995). Sensing SWC into an ion rich soil can be best performed using a thermally compensated sensor, high-frequencies and SWC calculation solely based on the dielectric permittivity real part $\varepsilon_R$.

Even if the soil type theoretically allows for the use of factory-calibrated sensors, the soil dielectric permittivity may significantly be affected by the soil texture and its organic matter content (Perdoc et al., 1996; Szypłowska et al., 2021). A soil-specific calibration may thus be required locally to adjust the coefficients of the manufacturer's transfer equation used for the determination of SWC based on the soil's dielectric permittivity. Several studies have shown the benefits of a soil-specific calibration for several soil types, including clayey soil. Different calibration methodologies are described:

- "Soil-suggested" calibration, which is based on an empirical function adapted according to soil texture, granulometry, acidity, organic matter content or even temperature. Such "soil-suggested" calibrations improve accuracy to a limited extent, however (Lukanu, and Savage, 2006).

- "*In situ*" calibration, which establishes the relation between the SWC estimated *in situ* with a factory-calibrated sensor and the actual SWC determined in lab by soil sample weighing, 1 point at a time. For example, Varble and Chávez (2011) show that every individual sensor position requires recalibration.

Furthermore, Dong et al. (2020) report laboratory simulations for *in situ* superficial soil sensor check; Jackisch et al. (2020) present *in situ* cross studies comparing several sensors; De Vos et al. (2021) perform true *in situ* calibration in forest soil and stress that soil samples should be collected periodically over a long period of time to cover the whole range of soil conditions (several years). This method is probably the most accurate, as the sensors are calibrated in real operating conditions. However, in our case, it was not possible to collect relatively dry clayey soil samples from deep layers.

The two cultivated plots where we conducted this study are located in the South West of France and are part of the ICOS ERIC network (Integrated Carbon Observation System, European Research and Infrastructure Consortium), whose member ecosystem stations must continuously monitor SWC at several depths. To this aim, FDR sensors are among the possible instruments and are implemented according to a standardized ICOS protocol on several ICOS ecosystem stations with various soil properties; our plots' soil is mainly clayey. The ICOS mandatory quality standards require sensor accuracy of at least 0.05 ($m^3m^{-3}$) over the whole expected SWC range. We questioned the relevance and accuracy of the factory-calibrated transfer functions because of the high clay content and the very heterogeneous characteristics of the soil in our plots throughout the year.

The objective of the present study is threefold: (1) to evaluate the accuracy of commercial FDR sensors on a clayey soil, using either the generic calibration constants provided by the manufacturer (raw SWC), or the specific soil calibration constants; (2) to compare SWC estimates based on either the dielectric permittivity modulus or on the dielectric permittivity real part and (3) to propose a FDR sensor soil-specific laboratory calibration process particularly suitable for clayey soils.

**2) Theory on dielectric permittivity-based techniques**


The dielectric permittivity $\varepsilon$, as expressed in the electromagnetics law, is measured in Faraday per meter (F/m) and formally the term "permittivity" is used only for "absolute permittivity". It can be expressed as a factor of "relative permittivity", denoted by "$\varepsilon_r$", and vacuum absolute permittivity, denoted by "$\varepsilon_0$": $\varepsilon = \varepsilon_r \varepsilon_0$

Most, if not all, sensors deliver a relative permittivity and most publications, including the present one, refer to

"dielectric permittivity" as a unitless number, which is actually the "relative permittivity".

Another important point is that the sensing of dielectric permittivity is carried out by processing an alternating AC signal which results in complex numbers formalism. Dielectric permittivity is therefore a complex number with a "real part" $\varepsilon_R$ (with a capital R, as opposed to the lowercase r of relative permittivity) and an "imaginary part" $\varepsilon_I$. The "modulus" $|\varepsilon|$ is the square root of the sum of squared real and imaginary parts:

$$|\varepsilon| = \sqrt{\varepsilon_R{}^2 + \varepsilon_I{}^2} \qquad\qquad\qquad \text{(Eq. 1)}$$

Most FDR sensors detect changes in the relative dielectric permittivity modulus.

Two main techniques have gained widespread acceptance for soil water content measurements: FDR (Skierucha and Wilczek, 2010), and TDR (Ledieu, 1986, K. Norobio, 1993). Both techniques rely on linking the soil dielectric permittivity measurement $\varepsilon$ to the soil water content (Color and Ulaby, 1974). FDR sensors detect the soil's

capacitance, which is its ability to store an electric charge, and is directly related to the soil dielectric permittivity. A maximum resonant frequency in the electrical circuit including the soil between the sensor's rods is determined and allows to estimate the water content. In the case of TDR sensors, a high frequency electric pulse is applied to the sensor rods inserted into the soil, travels the rods and is reflected by the rods ends. The measured travel time depends on the dielectric permittivity of the soil. Using a frequency range instead of a single frequency improves accuracy (by

mitigating the salinity bias, as discussed below).

As a first approximation, a linear relationship between the squared real part of the relative dielectric permittivity and the water content may be used.

$$\sqrt{\varepsilon_R} = A\theta + B$$

(Eq. 2)

With *A* and *B* being constants usually depending only on soil texture.

And consequently, the volumetric soil moisture linearity with $\sqrt{\varepsilon_R}$.

$$\theta = A_S\sqrt{\varepsilon_R} + B_S$$

(Eq. 3)

With $A_S$ and $B_S$ being constants ($A_S = A^{-1}$, $B_S = -BA^{-1}$)

Since at our study sites, FR-Lam and FR-Aur, soils are clayey and rich in ions, it is important to work with the real part of dielectric permittivity instead of the modulus of the dielectric permittivity, in order to avoid error caused by dielectric dispersion and the resulting resistive loss that mainly affects the imaginary part. Commercial FDR or other SWC sensors based on the real part of the dielectric permittivity are rare. Hence, one should pay attention to the sensors' specifications before installing them on site, especially for clayey soils.

A transfer equation (Eq. 2) is applied by commercial SWC sensors, either with factory fixed coefficients that cannot be changed or with resettable coefficients. In both cases, by post-processing correction or by reconfiguring sensor coefficients, it is possible to recover a more accurate estimate of SWC. Depending on the accuracy required for the

SWC measurements, it may be necessary to perform a soil-specific calibration, not only for each particular plot but even for each particular pit and depth, as on our stations or in the forest (De Vos et al., 2021).

**3) Material and methods**

**3.1) Soil description**

We carried out our study on two cropland ICOS stations (Fig. 1) in a very clayey region of south-western France: FR-Lam (43°29'47.21"N, 1°14'16.36"E), whose texture corresponds to silty-clay definition : 50.3% clay, mainly Kaolinite, 35.8% silt, 11.2% sand, 2.8% organic matter (Malterre and Alabert, 1963), and FR-Aur (43°32'58.80"N,

1° 6'22.01"E), which is also defined as a silty-clay soil: 30.8% of clay, mainly Smectite and Montmorillonite, 48.3% silt, 19.2% sands, 1.6% organic matter (Table 1). Both sites are certified in the ICOS network which means that their instrumentation and measurement protocols meet the required quality standards. For instance, on both sites, we installed HydraProbe (Stevens water monitoring systems Inc.) which are digital FDR sensors (Table 2). The accuracy required for ICOS sites is specified in the soil-meteorological measurement protocol (Op de Beck et al., 2018). It is

recommended to use FDR or TDR sensors with at least 0.05 $m^3m^{-3}$ accuracy over the entire SWC range. According to the manufacturer, the FDR digital sensor HydraProbe meets ICOS quality standards; the purpose of this study is to validate its accuracy on our study sites soil. In a preliminary check, both study plots showed a significant discrepancy between the factory-calibrated sensor SWC estimates and actual SWC (determined by weighing and measuring soil samples to calculate the soil volumetric water content). Therefore, we performed a thorough soil calibration on FR-

Lam and FR-Aur, into each pit and each depth. Here, we present only the results from the FR-Aur site as the collection of the samples was carried out later than on FR-Lam, and therefore with the benefit of hindsight and experience of the required delicate handling. So, no sample was damaged during collection or drying.

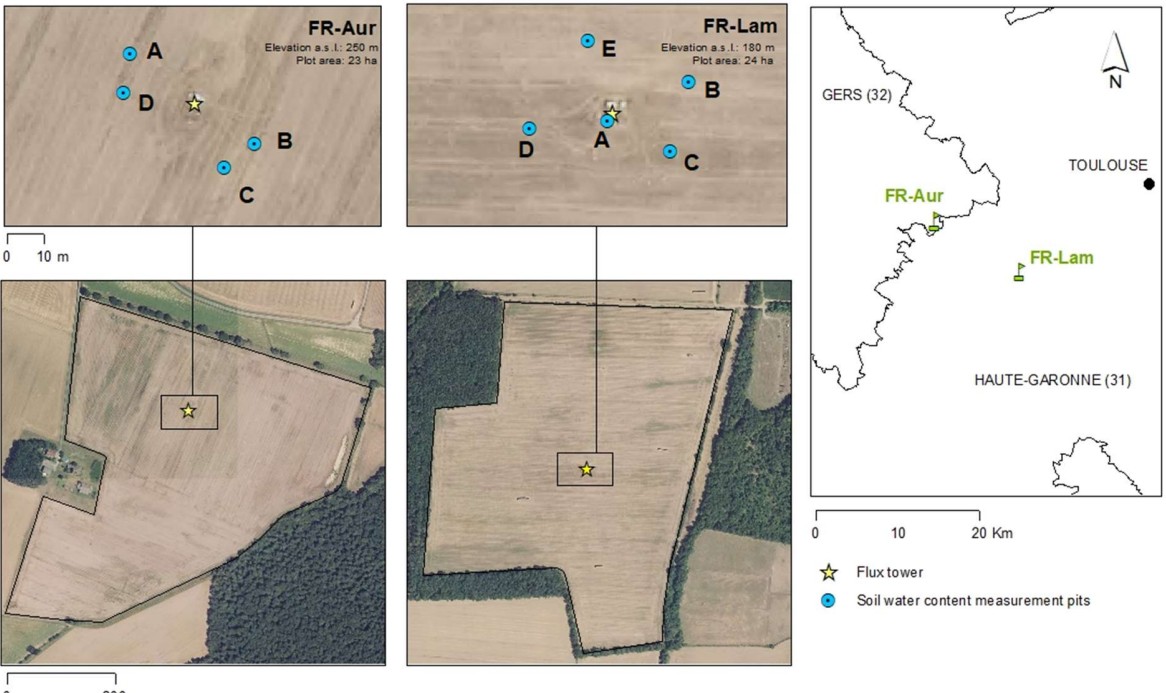

**Figure 1. FR-Aur and FR-Lam ICOS stations (https://www.icos-cp.eu/) and pit emplacements.**

**Table 1. Pit A FR-Aur soil contents**

| Depth range (cm) | Sand (% of mineral) | Silt (% of mineral) | Clay (% of mineral) | Organic C content (% of total) |
|---|---|---|---|---|
| 0-15 | 19.2 | 50 | 30.8 | 9.12 |
| 15-30 | 20.6 | 47.1 | 32.3 | 8.05 |
| 30-60 | 12.7 | 41.7 | 45.6 | 3.16 |
| 60-100 | 11.6 | 35.2 | 53.2 | 2.91 |

### 3.2) Soil sampling

On both sites, we collected soil samples at different depths (0-10 or surface, 5, 10, 30, 50, and 100 cm depth) into 4 to 5 pits, depending on the sites, to meet the ICOS requirements for SWC measurements (Op de Beek et al., 2018). When we created the 100 cm deep pits to install the HydraProbe sensors, we took the opportunity to collect soil samples at the depths required by the standardized ICOS protocol. Soil samples should be as wet as possible (at field capacity) when collected from the study site, so that the calibration process performed during in-lab drying covers the whole range of SWC. Clayey soil is probably one of the most difficult soils to handle. Undisturbed clayey soil sampling is difficult. In case of low SWC, sensor probes insertion or withdrawal into clayey soil may be destructive for the soil samples or the sensor rods. In order to minimize disturbance of the soil density during sample collection, we designed a homemade sample extruder (see Fig. 2). This apparatus is based on stainless-steel short tubes (soil sampler) of 70 mm internal diameter sharpened at the bottom, forced into the soil using a 5J perforator holding a sampler cloche. The collected soil sample volume is about twice as big as the sensed volume, which is less than 50 cm³. Fig. 2. (b) shows a sectional drawing of the sampler. To minimize soil compaction when the extractor is forced into the ground, this extractor was designed with two particularities. First, its tip is sharpened from the inner diameter to the outer diameter to mainly compact the remaining soil outside the soil sample. Second, the inner diameter at the sharpened edge is slightly smaller than the core sampler inner diameter to minimize the frictions between the soil sample and the inner sampler surface.

The soil sampler was forced horizontally into each pit at each required depth except for the soil surface, where it was forced vertically as the surface SWC sensors are also placed vertically. Once the tube is pushed all the way into the soil and extracted with the soil sample inside, a hydraulic carjack allows the soil sample to be gently pushed out of the sampler. It is important to place a thin round shaped PTFE sheet between the soil sample and the extruder piston to prevent the soil sample from sticking. All samples were hermetically sealed in plastic buckets that can withstand oven drying at 105°C, which is the temperature required to dry soil samples at the end of the calibration process. Soil samples were collected in duplicate, in case of technical problems during the set-up of the experiment (see appendix A for details).

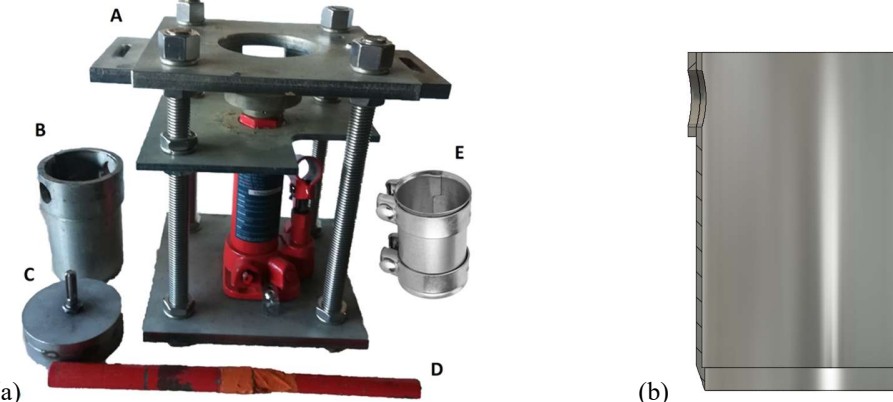

(a)                              (b)

**Figure 2. (a) Soil sample extruder: A) Carjack and stainless-steel frame, B) Soil sampler, C) Sampler cloche for pneumatic hammer, D) Carjack handle, E) Exhaust pipe clamp. (b) Sectional drawing of the sampler.**

### 3.3) Soil calibration

#### 3.3.1 Analog and Digital FDR sensors cross-calibrations

For the purpose of this study, we used several sensors and we assume that sensors of the same model are identical to each other. For practical reasons, we used an analog FDR sensor type with a cable that could be removed from the sensor's body for the weighing process, in order not to disturb the clayey sample by removing the sensor rods (Lukanu and Savage, 2006; see appendix A for details on soil calibration protocol and the different process steps). The analog FDR sensors were first cross-calibrated with the reference digital FDR HydraProbe sensor (SWC sensors' specifications are reported in Table 2). Indeed, only the reference digital FDR sensor provides the real part of the soil's relative dielectric permittivity $\varepsilon_R$. The analog FDR sensors have an analog output which is proportional to the internally calculated volumetric SWC value deduced from factory fixed coefficients. This way, we can also access the real part of the soil dielectric constant $\varepsilon_R$ using the analog FDR sensors.

During the cross-calibration step, both sensors were first placed in a large bucket with their rods into water-saturated clayey soil from our study site. The sensor bodies were covered with sand to slow down the evaporation, in order to limit crack formation and thermalize the sensors (Fig. 3a). We repeated this manipulation using sand as a substrate instead of clay to compare the intercalibration results. Figure 3b is a graphic comparison of the square root of the real part of $\varepsilon_R$, indicated by the digital FDR sensor, with the θ value (in V) indicated by the analog sensor. Experimental points can be best fitted to a second-degree polynomial regression. The second-degree polynomial equations were 190 similar whatever the substrate, clayey or sandy soil, showing a relatively low sensitivity of the cross-calibration to the soil texture in our case (data not shown). The obtained equation was used for subsequent $\sqrt{\varepsilon_R}$ deductions from analog sensors volumetric SWC sensing. Note that for future potential application by the scientific community, the cross-calibration should be carried out on soil from the study plot as the analog and digital sensors may behave differently in other soils types.

**Table 2. SWC sensors specifications**

| SWC sensor Model | Manufacturer | Output | Sensing base | Sensing range $(m^3m^{-3})$ | Rod length (cm) | Sampling volume $(cm^3)$ | Temperature sensing-compensation |
|---|---|---|---|---|---|---|---|
| HydraProbe, referred to as "digital" sensor | Stevens, 12067 NE Gleen Wilding Rd, Suite 106 Portland, Oregon 97220, USA | Digital (SDI 12) | The real part of the permittivity at 50 MHz | 0 - 1 | 4.5 | ≈40 | yes |
| DC2300, referred to as "analog" sensor | Beijing Dingtek Technology Co., Ltd. Room A209, Flounder Business Park, Shunbai Road 12, Chaoyang District, Beijing, 100022, China. | Analog (1-5V) | The modulus of the permittivity at 50 MHz | 0 - 0.6 | 6 | ≈45 | yes |


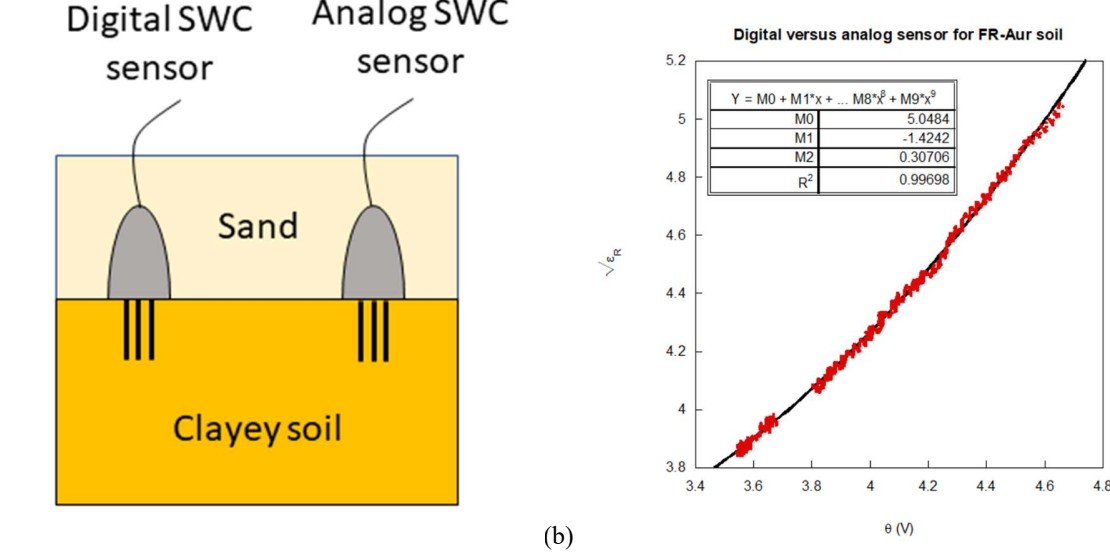

**Figure 3. (a) Analog and digital FDR sensors cross-calibration configuration (b) Graphic comparison of the square root of the real part of relative dielectric permittivity, indicated by the digital FDR sensor, versus the SWC θ value (in V) indicated by the analog FDR sensor. The table indicates the second-degree polynomial regression and the corresponding determination coefficient R².**

*3.3.2 Volumetric SWC estimation and associated error*

In this section, we detail our protocol for SWC estimation during sample drying. We calculated the actual volumetric soil water content (referred to as "real SWC") by weighing and measuring the soil samples, while simultaneously monitoring SWC obtained with the sensors, in order to compare the respective errors on SWC estimates, depending on sensor calibration strategy.

To determine the "real SWC", on a daily basis, on working days, we performed scale-based gravimetric measurements (EMS 12K0.1 scale, KERN & SOHN GmbH, Ziegelei 1, D-72336 Balingen-Frommern, Germany) of the slowly drying soil sample by subtracting the masses of the oven-dried soil sample, of the bucket, and of the sensor. The soil sample volume was also monitored using a generic digital caliper as the clayey soil volume may change (shrinking in so-called vertisol) (See appendix A for more details). Simultaneously, SWC values indicated by the analog FDR sensor were recorded by a data logger (CR1000, Campbell Scientific, Logan, Utah, USA). We proceeded by measurement of all samples from a particular pit at the same time, which means 6 samples (six depths) at once using 6 analog FDR sensors, until all of the 6 samples were completely dry, ensuring the whole SWC range was covered. Then, we repeated the operations for each of the 9 pits of our two study sites. In our case, the total soil calibration took 8 months.

For each sample, a second-order polynomial fit provides us with the transfer function between the sensor determined $\sqrt{\varepsilon_R}$ and the real volumetric SWC. It should be noted that a second-order polynomial fit (R²=0.997) was used instead of a linear regression (R²=0.989) to improve the accuracy of the modeling (see section 4.1).

Next, the relative errors on SWC estimate using factory calibration parameters of the FDR sensors were calculated using Eq. 4, where *FDR measured SWC* is the SWC estimated with the analog FDR sensor with its factory settings (transfer function to convert voltage signal into SWC) and *Real Volumetric SWC* is the SWC estimated with the gravimetric measurements.

$$Relative\ SWC\ Error = \frac{FDR\ measured\ SWC - Rea\ Volumetric\ SWC}{Real\ Volumetric\ SWC} \tag{4}$$

We used the determination coefficient (R², Eq. 5.) to compare the respective accuracies of calibration strategies.

$$R^2 = \frac{sum\ of\ squared\ regression\ (SSR)}{total\ sum\ of\ squares\ (SST)} \qquad (5)$$

230

## 4) Results and discussion

### 4.1) Vertisol issues

The FDR and TDR sensors provide volumetric sensing of the soil water content, not gravimetric water content, which is not the most adapted technique to estimate soil water content for vertisol (Zawilski, 2022). Indeed, vertisol specific shrinkage makes it difficult to accurately monitor drying soil sample volume, and micro and macro crack formation induce local errors. Vertisol shrinkage may be anisotropic (Mishra et al., 2020), so that measuring the height of the samples may not exactly reflect volume changes. However, as it is difficult to accurately measure the soil sample diameter inside the bucket, we considered shrinkage to be isotropic over the studied soil moisture range. This approximation is close to reality since the sample is not diametrically constrained and, with the exception of the bottom, air-surrounded. Concerning the issue of crack formation, it should be noted that the volumetric water content is the volume of the water contained in a soil sample divided by the total soil volume, including cracks. Hence, any SWC sensing technique is extremely dependent on soil spatial heterogeneity. On a shrinking soil (like clayey soil), cracks are often larger than the SWC sensor diameter, possibly introducing biases. Using multiple sensors may help mitigate errors, but crack formation is clearly a limitation to the use of FDR and TDR sensors in vertisol. Figure 4 shows the typical behavior of $\sqrt{\varepsilon_R}$ versus $\theta$ ("real volumetric SWC", determined by weighing and measuring soil samples). When the soil samples are progressively drying, the measurement curves are quite linear up to the point where crack formation begins. Then, the slope changes abruptly, becoming significantly steeper. To improve the modeling accuracy, second-order polynomial fits of squared relative dielectric permittivity real part were used for each depth and each profile. The linearity, at least before the crack's apparition, confirms isotropic shrinking assumption validity for the soil sample volume calculations.

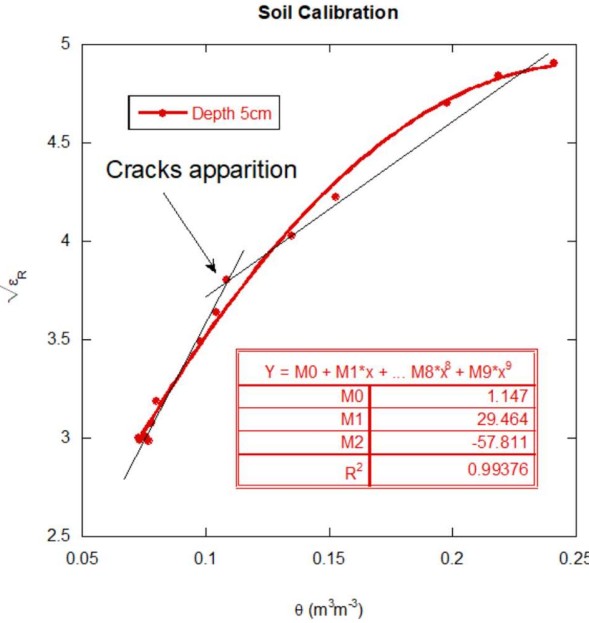

**Figure 4. Typical calibration relating the square root of the real part of relative dielectric permittivity to the "real volumetric SWC" (FR-Aur, pit D, depth 5 cm). For the purpose of this study, second-order polynomial regression was used for calibration (in red).**

**4.2) FDR sensor SWC estimation with factory-calibration and associated error**

*Estimated SWC versus "real volumetric SWC"* - We compared the real SWC (determined by weighing and measuring soil samples) with the SWC estimated using digital and analog FDR sensors, with the application of either factory calibrated coefficients or soil specific coefficients. Figure 5 displays the SWC estimated with the factory-calibrated FDR sensors versus real SWC measured at six depths, from surface to 100 cm, into pit A at FR-Aur site (see Fig. 1). For both FDR sensors, we found a large discrepancy between both methodologies of SWC measurements with a significant positive offset whatever the real SWC. This results in a significant overestimation of SWC using factory-calibrated FDR sensors. The overestimation was on average the highest ($0.10$ m$^3$ m$^{-3}$) for real SWC lower than $0.2$ m$^3$ m$^{-3}$.

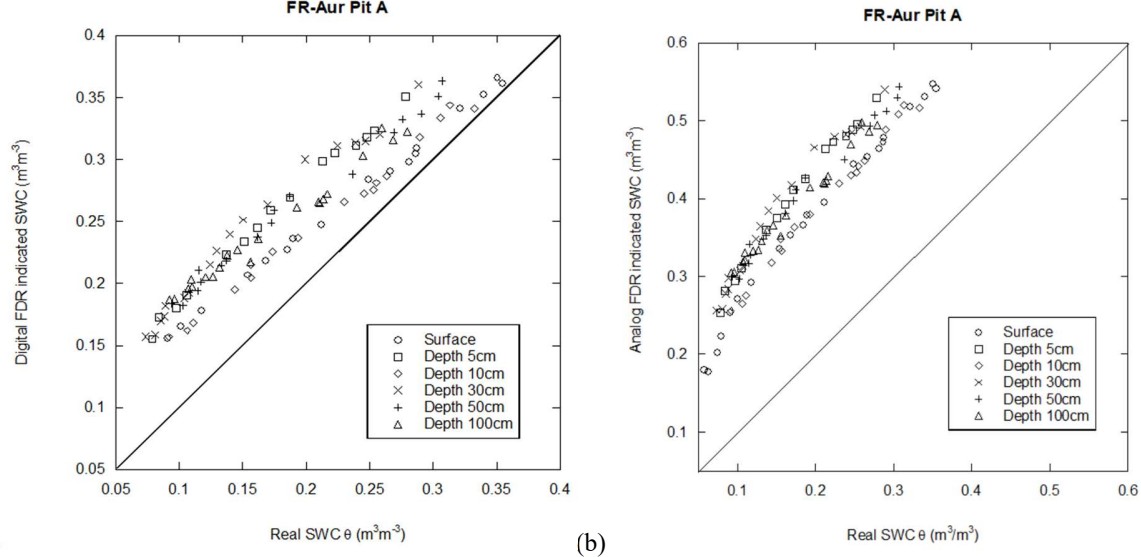

(a)  (b)

**Figure 5. SWC estimated with a factory-calibrated digital (a) or analog (b) FDR sensor, versus real SWC, at several depths into Pit A at FR-Aur site.**

*Relative error of estimated SWC versus real SWC* - Figure 6 shows the dynamics of the relative error (see equation 3 for definition) of the SWC estimated with the factory-calibrated digital (Fig. 6a) and analog (Fig. 6b) FDR sensors versus the real SWC measured at 6 depths into pit A at FR-Aur site. For both FDR sensors, the relative error decreased with increasing real SWC (from dry, $0.07$ m$^3$m$^{-3}$, to nearly water-saturated soil, $0.35$ m$^3$m$^{-3}$): from 115% to 1% with the digital FDR sensor and from 245% to 50 % with the analog FDR sensor. There was also a large scatter depending on the depth and the pit. Figure 7 displays relative SWC errors at depth of 100 cm into all four pits (see Fig. 1). We may note that pit A and B or pit C and D show similar relative error behaviors. However, between these two groups, the relative error gap is about 20% at the depth of 100 cm. For both FDR sensors and whatever the pit or the depth, errors were significant and positive, which means the soil was actually drier than the factory-calibrated FDR sensors would indicate. The drier the soil, the greater the relative error. The accuracy is way lower than required by the ICOS quality standards ($0.05$ m$^3$m$^{-3}$). It should be noted that the SWC derived from manufacturer's calibrations were so erroneous that the corresponding coefficient of determinations may be negative (Table 3).

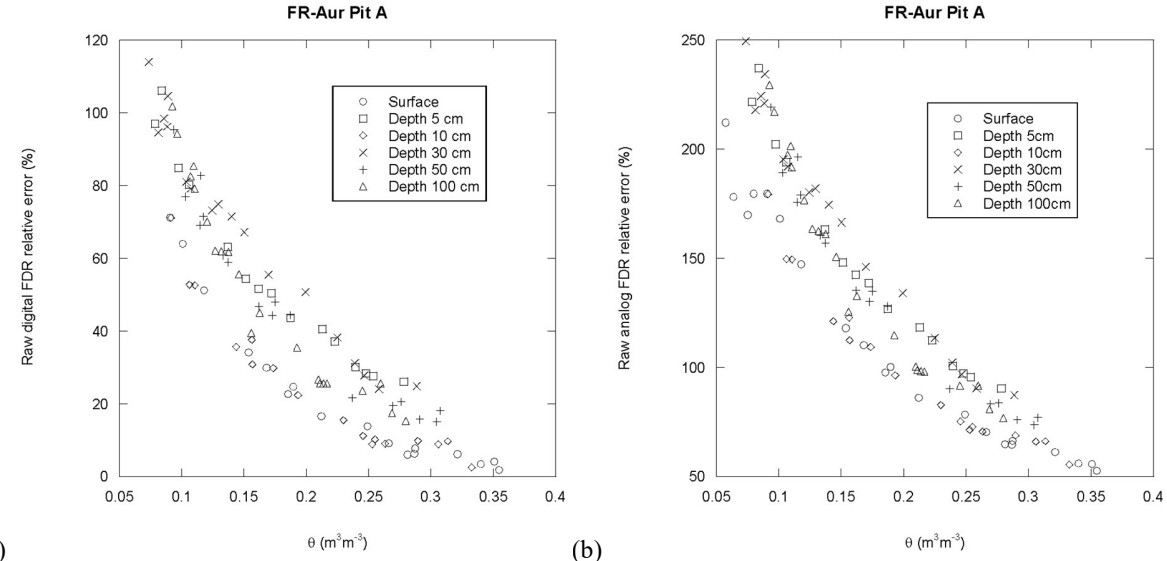

(a)                                                                                    (b)

**Figure 6.  Relative error of SWC estimated (a) with a factory-calibrated digital sensor based on the real part of the dielectric permittivity or (b) with a factory-calibrated analog sensor based on the modulus of the dielectric permittivity, versus real SWC into pit A at six depths.**

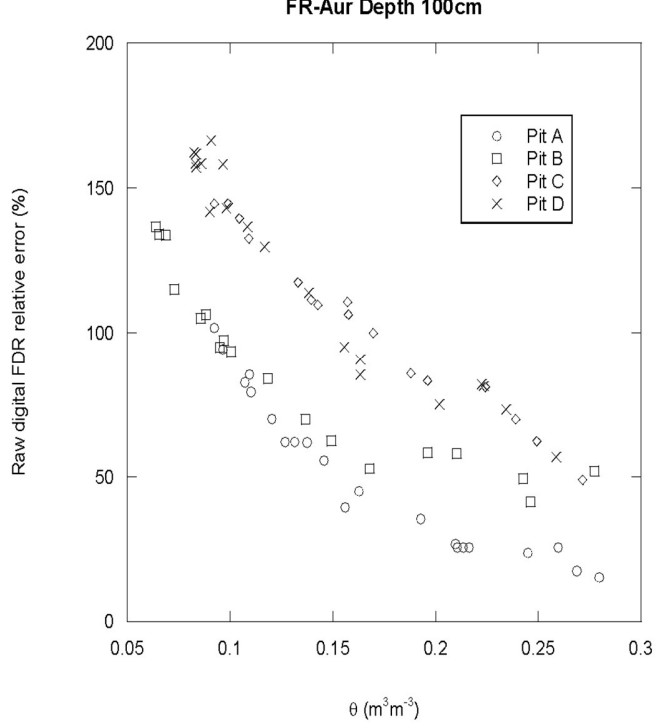

**Figure 7. Relative error on SWC estimated with factory-calibrated digital sensor versus real SWC for depth of 100 cm inside all four pits.**

**Table 3. Fr-Aur, pit A: relative errors and coefficient of determination before and after calibration**

| Depth (cm) | Digital sensor | | | | | | Analog sensor | |
|---|---|---|---|---|---|---|---|---|
| | Factory calibration | | Soil-specific $\varepsilon_R$ based calibration | | Soil-specific $\lvert\varepsilon\rvert$ based calibration | | Factory calibration | |
| | Relative error (%) at $\theta = 0.25$ $(m^3 m^{-3})$ | $R^2$ | Relative error (%) at $\theta = 0.25 (m^3 m^{-3})$ | $R^2$ | Relative error (%) at $\theta = 0.25 (m^3 m^{-3})$ | $R^2$ | Relative error (%) at $\theta = 0.25 (m^3 m^{-3})$ | $R^2$ |
| Surface | 21 | 0.80 | 1.7 | 0.996 | 0.9 | 0.996 | 78 | -2.1 |
| 5 | 42 | -0.19 | -4.2 | 0.991 | -2.0 | 0.995 | 96 | -11 |
| 10 | 21 | 0.70 | -3.3 | 0.985 | -3.2 | 0.992 | 73 | -5.0 |
| 30 | 48 | -0.60 | -5.6 | 0.987 | -5.5 | 0.987 | 94 | -10 |
| 50 | 37 | 0.02 | -4.8 | 0.989 | -4.6 | 0.989 | 88 | -7.8 |
| 100 | 30 | -0.53 | 2.4 | 0.985 | 1.0 | 0.986 | 87 | -12 |

*Modulus versus real part of the dielectric permittivity* - The relative error of SWC estimated with factory-calibration is around twice greater when using analog FDR sensors than when using digital FDR sensors (Fig. 6). This may be explained by their different operational modes. Indeed, the estimation of SWC with the digital FDR sensors is based solely on the real part of the dielectric permittivity, while the estimation of SWC with the analog sensors used for the in-lab calibration relies on the modulus of the dielectric permittivity. Soil ions affect mainly the imaginary part of the dielectric permittivity, which is in turn reflected in the modulus of the dielectric permittivity. Thus, the important shift observed in our study may not only result from the inadequate factory embedded calibration factors, but also from the high electric conductivity of the FR-Aur clayey soil. However, even if SWC estimate based on modulus was significantly improved after soil-specific calibration (Table 3), it should be noted that soil conductivity modifications, due to fertilization or liming for example, affect mainly the imaginary part of the dielectric permittivity and therefore the modulus of dielectric permittivity. Therefore, we found that the dielectric permittivity modulus-based sensors may be less accurate than the dielectric permittivity real part-based sensors. Soil conductivity changes with SWC; however, this variation was taken into account during the calibration. We would thus recommend the use of FDR sensors based on the real part of the permittivity for soils subject to large changes of electrical conductivity, such as cropland soils often submitted to fertilization operations.

**4.3) FDR sensor SWC measurements after soil-specific calibration**

Once soil calibration is performed, accurate coefficients can be applied to determine SWC based on the real part of dielectric permittivity. Figure 8 displays the same results as figure 5a after post-processing corrections with new soil-specific calibration coefficients for each pit and depth. Corrected digital FDR signals are much closer to the real SWC. After specific soil-calibration, relative error drastically decreases and the coefficient of determination is greater than $R^2 = 0.9$ (Table 3). For example, for a real SWC value of 0.25 m$^3$m$^{-3}$ at 30 cm depth, the *relative* error decreases to -5.6 and -5.5 % for estimated SWC with digital and analog FDR sensors respectively, with $R^2$ values of 0.987. Our experiment shows that soil-specific calibration in clayey soil allows for a dramatic improvement of the accuracy of SWC determination making corresponding errors well below 0.05 m$^3$m$^{-3}$ over the whole SWC range and depths. Hence, soil calibration ensures compliance with ICOS quality standards. The new calculated coefficient values after calibration vary significantly between pits for the same depth and between depths into the same pit, as shown in Fig. 6 and Fig. 7. In general, near the surface (from 0 to 10 cm) the calibration coefficients were more homogeneous and closer to the factory calibration coefficients than in deeper soil layers. This may be explained by soil homogenization by surface tillage, lower soil density (Namdar-Khojasteh et al., 2012) and lower clay content at the surface than in-depth.

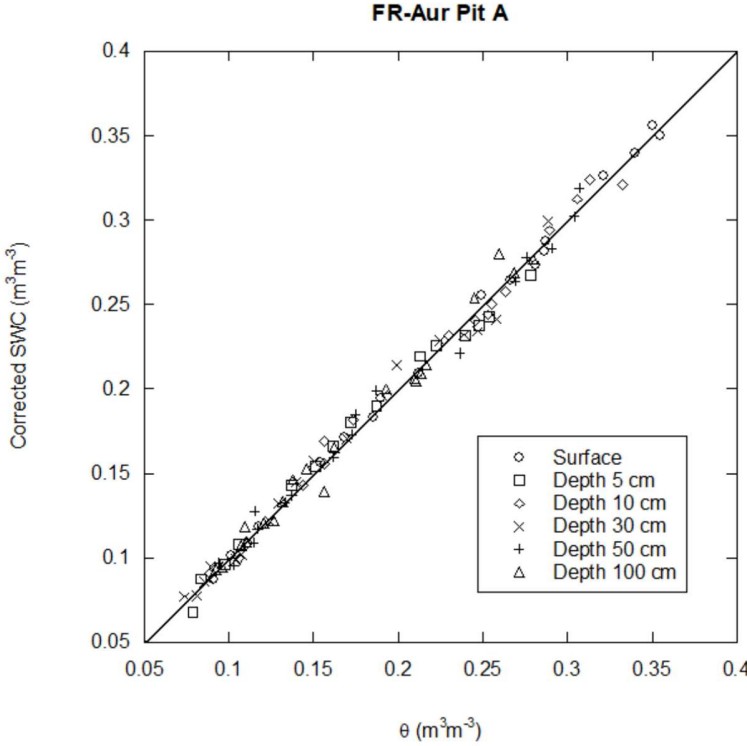

**Figure 8. SWC provided by digital FDR recalibrated sensor using soil-specific calibration coefficients for the same pit and depths as in figure 5a.**

**5) Conclusion**

This study highlights that using factory generic calibrations for SWC sensors with the same transfer function between permittivity constant and SWC, especially when they are based on dielectric permittivity sensing (FDR, TDR, Capacitance, radar, or microwave techniques), would not provide accurate estimates of SWC on every kind of soils. First, we demonstrated that the SWC relative error was clearly higher when using FDR sensors based on the modulus of the dielectric permittivity than on the real part of the dielectric permittivity. This was partly due to the high electric conductivity of our study site soils. Depending on the site soil and the field operations which in turn may affect the imaginary part of the dielectric permittivity and thus bias the estimated SWC, it is highly recommended to use FDR sensors based on the real part of the permittivity for cropland soils often subject to major fertilization operations. Secondly, we show that in the case of clayey soils, a laboratory calibration is needed to ensure accuracy of the soil water content determination. Indeed, we found that the sensing of dielectric permittivity to determine SWC in clayey soils is highly influenced by spatial heterogeneity in terms of texture, density and physicochemical properties. Without soil-specific calibration, we observed an increase of the relative error when the soil turns very dry. This relative error can reach up to 115% on cropland soils when using sensors based on the real part of the dielectric permittivity and up to 245% when using the sensors based on the modulus of the dielectric permittivity. We show that performing soil-specific calibration at a specific sensor location allows to adjust the constants of the transfer equation, ensuring very accurate SWC estimates at that specific location with FDR sensors based on the real part of the permittivity. After soil-specific calibrations, SWC calculations errors are well below 0.05 $m^3 m^{-3}$ over the whole SWC range and depths. We recommend to always check if the SWC is accurately determined with the factory-calibrated commercial sensors in the soil of interest before conducting studies such as the estimation of the extractable soil water and water reserve, the study of soil microbial processes, soil water and greenhouse gas fluxes, and/or characterization of their spatial variability. If accuracy is not sufficient, perform soil calibration at each specific location. Soil calibration is long and

manpower-consuming but may be necessary. It would be interesting to test our soil-calibration process with remote sensors, using satellites, which have the advantage to assess SWC without physical contact.

**Appendix A: Specific clayey soil calibration protocol**

**Setup**

- Soil samples should be as wet as possible when collected from the study site, so that the calibration process performed during in-lab drying covers the whole range of SWC. They should be large enough to accommodate the sensor rods. Soil sample should be collected in duplicate, in case of technical problems during the set-up of the experiment.

- Use a data logger, such as Campbell's CR1000, programmed for soil moisture sensor monitoring and wired with labeled cables for each labeled sensor.

- Use a scale with sufficient weighing capacity and resolution.

- Use buckets that are big enough for the soil samples and that can withstand a temperature of 105°C (polypropylene buckets are suitable).

- Use a caliper for measuring soil samples' dimensions.

- Use a stainless-steel exhaust pipe clamp, to hold the sample and prevent it from being altered during sensor rods' insertion of internal diameter and height fitting soil samples external diameter and height, to hold the sample and prevent it from being altered during sensor rods' insertion.

**Preliminary measurements**

Weigh each soil moisture sensor without its cables: $W_P$

Weigh the buckets: $W_B$ (may be used for uncertainty determination).

**During each measurement cycle:**

Sensors are inserted into the soil samples and placed individually inside a bucket.

Note: an exhaust pipe clamp of fitted dimensions can be used to hold the sample and prevent it from being altered during sensor rods' insertion.

Three points are marked on each soil sample around its circumference, every 120° (See Fig. A1, this will be necessary to determine the sample dimensions during the measurement cycle, by averaging).

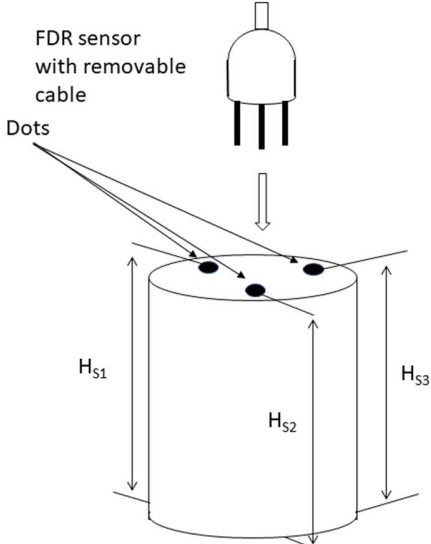

**375**

**Figure A1. Calibration setup**

Sensors are connected to a logger and surrounded by tissue paper to slow down the evaporation from the soil samples.

**Routine measurement**

**380**  On a daily basis, on working days:

-    Relative dielectric permittivity ($\varepsilon_R$) values are reported from the logger for each sensor.

-    The tissue paper surrounding the sensors is set aside and the sensor cables are disconnected before each sample is weighed (including bucket and inserted sensor): $W_{SBP}$.

-    The height of three points around the circumference of each soil sample are measured with a caliper: $H_{S1}$,
**385**  $H_{S2}$, and $H_{S3}$, along with the sample diameter $D_S$ (if possible), in dekameters (dam).

-    It is welcome to take a clear picture of the samples to track any apparent crack formation.

-    Sensors are reconnected and the tissue paper is put back in place.

**Once the soil samples are considered completely dry:**

-    When the measurement cycle is considered finished, it may be necessary to rewet the soil sample to
**390**  withdraw the sensor rods.

-    Each sample is dried in an oven at 105°C for two days, into its bucket but without the SWC sensor, before final weighing (including the bucket): $W_{SB}$.

### Data processing

The soil water content weight ($W_w$ = water weight, in kg) is calculated by subtracting the weight of completely dried soil sample (including bucket) ($W_{SB}$) and the sensor probe weight ($W_P$) to each daily soil sample weighing including the bucket and the inserted SWC sensor probe ($W_{SBP}$):

$$W_W = W_{SBP} - W_{SB} - W_P$$

(A1)

Note: With water density being constant and equal to 1 kg/liter, the water volume $V_W$ (in liters), present in the soil samples during the measurements, is numerically equal to the water mass (in kg):

$$V_W = W_W$$

With the samples height (and diameter if available) measurements, the soil samples volume is calculated (in liters):

$$V_S = \frac{(H_{S1} + H_{S2} + H_{S3})}{3} \pi (\frac{D_S}{2})^2$$


(A2)

Note: As an approximation, if the sample diameter ($D_S$) was not measured, due to the inaccessibility of the sample into the bucket, it should be estimated by assuming that it varies along with the mean of the three measured height (isotropic shrinkage).

We can then determine the samples' volumetric soil water content (SWC or θ) in m³/m³ (or in liters/liters)


$$\theta = V_W V_S^{-1}$$

(A3)

Lastly, plotting these values of the samples' volumetric soil water content (SWC or θ), based on sample weighing, on a graph versus the square root of the real part of $\varepsilon_R$, as indicated by the FDR sensor, enables us to infer the calibration constants of the sensor ($A_S$, $B_S$ and $C_S$), using the following regressions (whichever fits best):

$\theta = A_S\sqrt{\varepsilon_R} + B_S$ (linear fit)

$\theta = C_S\varepsilon_R + A_S\sqrt{\varepsilon_R} + B_S$ (second-order polynomial fit)

(A4)

### Authors' contributions

This study was conceptualized by BZ, who carried out a preliminary investigation showing the benefits of soil-specific calibration, developed the soil calibration methodology protocol, designed and built the soil sample extruder apparatus, participated in the soil samples extraction, measurements, and formal analysis, and wrote the first draft. FG was involved in the soil sample extraction. NC was involved in the measurements. AB reviewed the draft and participated in the measurements. TT was involved in the soil sample collection, measurements, formal analysis, writing and reviewing of the original manuscript.


**Code and data availability.**

The data and source code used for these studies can be obtained by contacting the author.

**Competing interests.**

The author declares that he has no conflict of interest.

**Financial support.**

This project was funded by the Institut National des Sciences de l'Univers (INSU) through the ICOS ERIC and the OSR SW observatory. Facilities and staff are funded and supported by the Observatory Midi-Pyrenean, the University Paul Sabatier of Toulouse 3, CNRS (Centre National de la Recherche Scientifique), CNES (Centre National d'Etude Spatial), INRAE (Institut National de Recherche pour l'Agronomique et Environnement) and IRD
(Institut de Recherche pour le Développement).

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
