# Peer review of "Calculation of soil water content using dielectric permittivity-based sensors; benefits of soil-specific calibration."

_Geoscientific Instrumentation, Methods and Data Systems, 2022_

## Referee Comment (RC1)

Review Report

**Calculation of soil water content using dielectric permittivity measurements; benefits of soil-specific calibration.**

**General remarks**

I consider this manuscript as a draft version which needs considerable further elaboration to become a well structured and informative paper. The measurement basis is good but the statistical analysis and results need to be presented and discussed more thoroughly.

I also advise to improve the English language considerably.

Some general suggestions for improvement:

- **Introduction** needs to be extended including more recent references and findings on this topic. At the end of the introduction readers expect clear objectives of the study, or a list of research questions. These need to be addressed in the Discussion and Conclusions.
- In **Material and Methods**, clearly describe the test sites and sampling procedure, and inform the reader of the type of soil and its clay and organic content by depth layer in a table. Also, clearly describe the laboratory setup and refer to the protocol in Appendix A
  Describe the statistical analysis, and introduce some measures for assessing bias and precision to evaluate the recalibration performance
- In **Results & Discussion section** use Tables to evaluate the performance after recalibration, but also to show the difference between using the real part or modulus of dielectric permittivity
- Provide in Discussion some answers to "What is acceptable accuracy of SWC measurements ?" and "What is the minimum set of replicate samples per depth needed for proper recalibration ?"
- In **Conclusion:** how are your research questions answered and objectives reached ?

Below you can find comments by Line, also for the numerous typo's. Hope these can help you to improve the MS.

**Introduction**

L21. Climatic "soil" variables – add "soil"

L26. Accuracy of 3%, do you mean 3% absolute (so volume %), or 3% relative compared to the gravimetrically determined SWC ?

L27. What do you mean by "points" ? Measurement locations ?

L29. "Roods" must be Probe "rods"

L37. Remove "usually", OM is never represented in a soil textural triangle

L49 both are dependent on linking accurate soil capacitance with soil water content

L50 traveling "through" the rods

L55 roods => "rods"

L71. choose the right probes for specific soils such as clayey soils => what are the specification of such rods ? Explained further ?

L74. To recover a more accurate measurement => to reach a more …

L75. but even for a particular pit and particular depth for accurate SWC measurements. => see paper: https://www.mdpi.com/2076-3417/11/24/11620

**M&M**

L79-85: I suggest to put the applied sensors in a table listing their characteristics (also rod lengths), as for example in Table 1 of  https://essd.copernicus.org/articles/12/683/2020/

L86-92. Nice 'home made apparatus" – Have you checked bulk density differences compared to conventional volumetric sampling ? is the sample really taken by pneumatic hammering ? Or just pneumatic forcing into the soil ? What if there is a stone content in the soil ? How do you cope with samples containing coarse fragments  ?

L95. So from Figure 2 I conclude you have 4 replicate SWC pits in FR_Aur and 5 in FR-Lam ? Please specify. Landuse is cropland ?

L 98. Refer to ICOS programme (https://www.icos-cp.eu/)

L105. As an illustration for this paper, some FR-Aur results are shown.Where ? Please refer to figure or table

L106. Why are samples taken differently in the topsoil (vertically) compared to the subsoil layers ? I would take them all horizontally for a study.

L105-109. Here you do not mention any hammering ? Just pressure.

L110 "was near water-saturated" – water saturation, but probably you mean the soil was at "field capacity".

L119.. List the properties of the analog and digital FDR probes in the instrument table please, so that the reader knows which devices these are.

L128. Cracking is indeed one of the biggest problems of FDR measurements in clayey soils. You can avoid it in the lab, but how to cope with this in the field (especially topsoil) ?

**Results and discussion**

L154 "Are the cack volume parts of the sample
volume?" – typo "Crack" – by convention dry bulk density is the ovendry mass of soil (dried at 105°C) devided by its volume when taken in the field, mostly at field capacity. So in this sense, when a clayey sample dries out and is cracking, the crack volume is part of the sample volume, and no substraction is needed.  Therefore it is called "bulk density", because it also includes pores, and channels, and cracks … in contrast to specific density of soil.

L160. Figure 4. It would be informative to show progressive crack formation upon drying and this SWC- diel. Permittivity relationship.

L165-167. This is probably linked with the fact that the tested soil samples originate from arable land that is homogenised by ploughing. In forest and permanent grassland soils topsoil variability is usually greater and less homogenous.

L170.(and L182) Figure 5 clearly shows overestimation of SWC by FDR, which has been reported by quite a lot of studies (e.g. https://doi.org/10.1016/j.agwat.2011.09.007)

L195. How the relative error is calculated should be part of M&M section. Not in results.

L202. The dryer is the soil and the greater is the relative error.

L213. Once soil calibration is done, new calibration constants can be injected into the relations between SWC and the real
part of dielectric permittivity.. Clearly show how the recalibration is effectively performed, i.e; ; how soil-specific calibration coefficients are determined.

**CONCLUSION**

Refer to objectives of this study and answer the research questions.

**Appendix A**

L242. "calibration process is made during sample drying." For clayey soils there is a hysteresis effect. Is calibration different when using the drying path compared to the rewetting trajectory ?

L258. Why are you not taking calibrated digital photo's to estimate the dimensions by digital image processing ? You are already taking photos for the cracks (L270)

L221 Papet => "Paper"

**Some other reflections**

L38. *a soil-specific calibration may be required locally to determine the proper calibration of moisture versus dielectric permittivity constants*
Is it better to do soil-specific calibration directly on the dielectric permittivity response of the sensors (as you did) or -in case of FDR, on derived sensor output signals like "period average" (or travel time in TDR) which also includes sensor characteristics ?

---

## Referee Comment (RC2)

[referee-annotated manuscript omitted]

---

## Author Comment (AC1)

**Responses to reviewers**

**RC1 comments:**

We are sincerely grateful to our first referee hereafter called RC1 and to Steven Evett, hereafter called RC2, for taking their time to read, comment and help to improve our manuscript.

RC1: "Introduction needs to be extended including more recent references and findings on this topic. At the end of the introduction readers expect clear objectives of the study, or a list of research questions. These need to be addressed in the Discussion and Conclusions."

References were added. The objectives have been clearly stated at the end of the introduction and addressed in the discussion and conclusion sections.

RC1: "In **Material and Methods**, clearly describe the test sites and sampling procedure, and inform the reader of the type of soil and its clay and organic content by depth layer in a table. Also, clearly describe the laboratory setup and refer to the protocol in Appendix A. Describe the statistical analysis, and introduce some measures for assessing bias and precision to evaluate the recalibration performance".

The M&M section was rewritten, reorganized and enriched with the requested information. We do not have soil analyses for each pit and depth. However, we included table 1 with the soil data that we have for FR-Aur site only. It is beyond the scope of this paper to establish a correspondence between soil composition, soil texture or soil density, and the calibration constants. The scope of this paper is only to suggest a check and soil-specific calibration using the soil of each planned SWC sensor implementation. Thereby, we do not need to know why the commercial sensors with the factory calibration constants overestimate or underestimate the real SWC. As it is rather difficult to work with clayey soil, we are suggesting a protocol that allows us to successfully conduct a calibration campaign. The introduction and conclusion were clarified.

The quality levels of the calibrations were evaluated using the determination coefficient between soil-specific calibrated SWC estimation and Real SWC ($R^2$, see Eq. 4.). Then, the error made on SWC estimation relative to real SWC with the use of analog or digital FDR sensors factory settings was calculated using Eq. 3. There is not one emplacement where a factory-calibrated sensor is providing SWC within ICOS mandatory 0.05 $m^3/m^3$ accuracy. Different locations imply different calibration coefficients, which may be close to each other or not, unpredictable before the actual local calibration.

RC1: "In **Results & Discussion section** use Tables to evaluate the performance after recalibration, but also to show the difference between using the real part or modulus of dielectric permittivity

We added table 3 to support the performance of FDR sensors after recalibration. We also extended the results to a comparison between real part and modulus of the permittivity-based sensing.

RC1: "Provide in Discussion some answers to "What is acceptable accuracy of SWC measurements?" and "What is the minimum set of replicate samples per depth needed for proper recalibration?"

According to the ICOS protocol (Op de Beeck and al. reference added), the SWC probe should have at least 0.05 ($m^3/m^3$) accuracy over the whole expected SWC range sensed into four pits. We added this information inside the introduction and Mat & Meth sections (see lines 68 and 129).

RC1: "In **Conclusion:** how are your research questions answered and objectives reached ?"

We have rewritten the conclusion to reflect more clearly how the objectives of our research were met.

RC1: "Below you can find comments by Line, also for the numerous typo's. Hope these can help you to improve the MS."

We are grateful to RC1 for his suggestions and we modified our manuscript in consequence.

RC1: "L26. Accuracy of 3%, do you mean 3% absolute (so volume %), or 3% relative compared to the gravimetrically determined SWC ?"

The value of "3%" is coming from the SWC sensors manufacturer's manuals and some of them, but not all, present that this is an absolute error of 0.03 $m^3/m^3$. We modified this value to the "0.03 $m^3/m^3$" value (see line 32)

RC1: "L27. What do you mean by "points" ? Measurement locations ?"

"points" means "other soil features" cited in the next sentence. Text modified (see line 34)

RC1: "L37. Remove "usually", OM is never represented in a soil textural triangle"

We are aware that OM is never represented in a soil textural triangle. We modified the sentence to clarify our message (see line 46 ).

RC1: "L71. choose the right probes for specific soils such as clayey soils => what are the specification of such rods ? Explained further ? "

This point was raised in the previous sentence "ionic soil, such as clayey soil, requires a real part dielectric constant based probe." This point was developed for better clarity. (see lines 109-113)
Table 2 with sensor specifications was added.

RC1: "L75. but even for a particular pit and particular depth for accurate SWC measurements. => see paper: https://www.mdpi.com/2076-3417/11/24/11620"

This reference was added into the text.

RC1: "L79-85: I suggest to put the applied sensors in a table listing their characteristics (also rod lengths), as for example in Table 1 of https://essd.copernicus.org/articles/12/683/2020/"

Characteristics of sensors were added in table 2.

RC1: "L86-92. Nice 'home made apparatus' – Have you checked bulk density differences compared to conventional volumetric sampling ? is the sample really taken by pneumatic hammering ? Or just pneumatic forcing into the soil ? What if there is a stone content in the soil ? How do you cope with samples containing coarse fragments ?"

Point by point:

- Thank you.
- The conventional volumetric sampling we know is to enforce a collar of a known volume into the soil, withdraw it, and crush the contained soil to liberate it from the collar making it unusable for further SWC measurement. Both

methods use a collar or a sampler forced to the soil. The sampler forced into the soil may seem to be highly invasive and potentially compact the soil sample. However, the sampler was designed to minimize eventual compaction (a figure with the sampler sectional draw was added). Also, when the sampler was enforced to the soil, the soil sample surface inside the sampler was at an equal level as the surrounding soil surface which was plaid for compaction exemption.

- Sampler enforced with a 5j perforator (added to the text).

- Stones are present on the Fr-Aur station (a few percent), We have a chance to not encounter any stone problem, nor during sample collection or further calibration process. However, it was planned that in case of sample collection impossibility due to a stone to collect another adjacent sample. In the case of SWC sensor insertion to the soil sample impossibility, it was planned to use the second sample (two samples by location were withdrawn). Of course, it is not impossible that both samples contain stones preventing them to be used but this was highly improbable on FR-Aur.

- The aim of the soil sample collection and the further use for calibration was to work as close as possible to the real conditions. Consequently, we do not discard any coarse elements from the soil samples. As we work on clayey soil, the only coarse elements are the stones.

RC1: "L95. So from Figure 2 I conclude you have 4 replicate SWC pits in FR_Aur and 5 in FR-Lam ? Please specify. Landuse is cropland ?"

Correct, we have four pits on FR-Aur and 5 pits on Fr-Lam and both stations are cropland stations. Specifications were added to the text.

RC1: "L 98. Refer to ICOS programme (https://www.icos-cp.eu/)"

Reference added into the figure 1 (ex fig. 2) description.

RC1: "L105. As an illustration for this paper, some FR-Aur results are shown. Where ? Please refer to figure or table".

The whole text of this paper presents only the results of the FR-Aur soil calibration for more clarity. FR-Lam soil calibration is qualitatively comparable but not complete.

RC1: "L106. Why are samples taken differently in the topsoil (vertically) compared to the subsoil layers ? I would take them all horizontally for a study. "

Explanations were added to the text (see lines162). The "surface" sensors are vertically enforced to the soil and other depth sensors are placed horizontally so soil samples were collected according to the SWC sensors' placements.

RC1: "L105-109. Here you do not mention any hammering ? Just pressure."

Word "forced" was added to the text (see line 154).

RC1: " L110 "was near water-saturated" – water saturation, but probably you mean the soil was at "field capacity"."

The expression "at field capacity" was added to the text (see line 149). Please note that the "water-saturated soil" expression is also used.

RC1: "L119. List the properties of the analog and digital FDR probes in the instrument table please, so that the reader knows which devices these are."

See Table 2.

RC1: "L128. Cracking is indeed one of the biggest problems of FDR measurements in clayey soils. You can avoid it in the lab, but how to cope with this in the field (especially topsoil) ?"

When the circularly distributed rod probes are used (it is the case with our sensors) the macro cracks mainly form around the probes as the soil is maintained by the rods. However, micro-crack formation is not avoidable. Not only the cracks forming between the rods but also around the rods with consequent poor electric contact between the rods and the soil. To our knowledge, there is no "magic" solution for that.

RC1: "L154 "Are the cack volume parts of the sample volume?" – typo "Crack" – by convention dry bulk density is the oven dry mass of soil (dried at 105°C) devided by its volume when taken in the field, mostly at field capacity. So in this sense, when a clayey sample dries out and is cracking, the crack volume is part of the sample volume, and no substraction is needed. Therefore it is called "bulk density", because it also includes pores, and channels, and cracks … in contrast to specific density of soil."

Absolutely. Then volumetric water content is the volume of the water divided by the volume of the soil including its cracks, any SWC sensing is then extremely localization depending. On a shrinking soil like our station's soil, cracks are often larger than the SWC sensor diameter. In the case of the presence of the crack any relatively small volume sensing device such as FDR, TDR, and so on provide a biased SWC estimation. Also multiplying the sensors does not help a lot since these sensors need to be inserted into the soil and do not work when one or more rods are in a crack. This is clearly a limitation for FDR and TDR sensors use in vertisol. Sentences were added to the text (see section 4.1).

RC1: "L160. Figure 4. It would be informative to show progressive crack formation upon drying and this SWC-diel. Permittivity relationship."

Cracks formation observation is a challenge. There are three stages of the formation of the cracks: Vertical cracks formation visible on the surface, horizontal cracks formation inside the vertical cracks which are not visible from the surface, and vertical cracks formation inside the horizontal cracks which are not visible either. The link between the cracks and the permittivity would be a very interesting subject to study but it is definitely beyond the scope of our paper.

RC1: "This is probably linked with the fact that the tested soil samples originate from arable land that is homogenized by plowing. In forest and permanent grassland soils topsoil variability is usually greater and less homogenous."

Effectively, the apparent homogeneity of a surface soil is most probably linked with the soil tillage. The corresponding sentence was added (see lines 322).

RC1: " L170.(and L182) Figure 5 clearly shows overestimation of SWC by FDR, which has been reported by quite a lot of studies (e.g. https://doi.org/10.1016/j.agwat.2011.09.007)"

We agree that most of the studies showed an overestimation of the sensed SWC on clayey soil but not all. As the estimation error is determined by the factory-implemented calibration factors of the concerned sensors, everything is possible. Most of the sensors are factory calibrated for the most common soils and, with these settings, FDR sensors overestimate SWC assessments. However, the same sensors with different calibration constants may also underestimate SWC. It is just a question of settings.

RC1: "L195. How the relative error is calculated should be part of M&M section. Not in results."

Equation was moved to the M&M section (see line 228).

RC1: "L213. Once soil calibration is done, new calibration constants can be injected into the relations between SWC and the real part of dielectric permittivity. Clearly show how the recalibration is effectively performed, i.e; ; how soil-specific calibration coefficients are determined."

The specific calibration coefficients were determined by fitting the curves of "real soil water content versus indicated real part of the dielectric relative permittivity". Explanations were developed in M&M.

RC1: "L242. "calibration process is made during sample drying." For clayey soils, there is a hysteresis effect. Is calibration different when using the drying path compared to the rewetting trajectory ?"

The hysteresis is mainly due to the cracks' opening and closing. Cracks closing is favored by the internal soil pressure due to the surrounding soil. When a soil sample is used, there is no surrounding soil so the hysteresis observed with the soil samples is sensibly different from the hysteresis observed with real conditions. The rewetting was not attempted.

RC1: "L258. Why are you not taking calibrated digital photo's to estimate the dimensions by digital image processing ? You are already taking photos for the cracks (L270)".

Sample diameter determination would be possible by digital photo processing but seems to us inaccurate especially when the sample soil is shrinking. The sample height would be not available as the samples are bucket surrounded.

RC1: "L38. "*a soil-specific calibration may be required locally to determine the proper calibration of moisture versus dielectric permittivity constants*" Is it better to do soil-specific calibration directly on the dielectric permittivity response of the sensors (as you did) or -in case of FDR, on derived sensor output signals like "period average" (or travel time in TDR) which also includes sensor characteristics ?"

As always, it is preferable to calibrate the whole process including as many characteristics as possible for precise calibration of a *specific sensor*. This calibration would be precise for only one model of a sensor. In case of any change in the sensor characteristic, all the calibration processes must be redone. The soil calibration allows to link dielectric permittivity with the SWC and does not allow to get rid of the sensor imperfection but is still valid for a "correctly" working sensor. "Correctly", means that the soil dielectric permittivity measurement by this sensor is accurate enough.

In this study, we are assuming that the main problem with SWC sensing comes from the relation between the permittivity and the SWC as this relation is soil-dependent. Indeed, factory injected coefficients cannot be universal and soil-suggested calibration factors are not yet successful. However, a soil-specific calibration with the soil coming from the planned sensor emplacement, even if it is long, is still possible.

**RC2 comments:**

RC2: This paper was very difficult to read and understand. The English is somewhat fractured and word choice is sometimes inappropriate. I have added notes in several places in the first part of the manuscript PDF to help improve the text in this respect. I hope those guidelines will be followed for the rest of the text."

We are grateful to RC2 for his annotations for helping us with the text improvement. Our manuscript has been corrected by a native English-speaking person.

RC2: "Change "probe" and "probes" to "sensor" and "sensors". These are soil water sensors. A probe does not necessarily involve a sensor. "

The "probe(s)" words were changed when necessary to improve the readability of our paper. Please note that we kept Stevens SWC sensor's name "HydraProbe" and Delta-T SWC sensor's name "ThetaProbe".

RC2:" Change "measurement" to "sensing", "measure" to "sense", and so on. The sensors involved do not measure soil water content and they do not measure dielectric permittivity. They measure frequency and deduce permittivity and water content from that measurement – they are thus soil water content sensors."

The title and some expressions in our paper were reformulated according to RC2's suggestion. The new title is: "Calculation of soil water content using dielectric permittivity-based sensors; benefits of soil-specific calibration." However, it is even simplest because only the voltage can be *measured* and even the frequencies are only deduced. But this "language abuse" or "language shortcut" is widely used and accepted and when a sensor is indicating a value, that value is called "measurement" even if it is only a sensed deduction. This is the so-called *indirect* measurement. Formally speaking a thermometer does not measure the temperature, a digital caliper does not measure the length, a scale does not measure the weight, and so on. Nota bene, a scale is sensing the weight (force) whose unit is not "Kg" but "N" and the weight is only proportional to the mass with a non-constant factor depending on the air pressure or altitude. A soil sample has the same mass on the Earth and the ISS station but not the same weight. So that when "weighing", formally, it is incorrect to note it in "kg". However, in the current language all these sensors "measure" the corresponding values. For example, the title of one of the numerous publications of RC2 is: "Soil Water Measurement by Time Domain Reflectometry".

RC2: "Write "capacitance" not "capacity".

Done, effectively "capacity" is not the correct word

RC2: "The graphs are not easy to understand because the colored symbols are too small, and too similar in color in some instances. Please use different symbols for the different depths and use black and white, not colors. Colors are particularly difficult for the color blind."

We are sensitive to that remark and we have done some changes to improve the readability of graphs. Making the symbols bigger may make it difficult to see that the sensed soil water content curves are far from the real soil water content and also not superimposed.

RC2: "The authors took samples from pits in soils containing smectitic/montmoronillitic clays and from pits in soils containing kaolinitic clays. This is potentially quite interesting because it is known that these clay types have much different cation exchange capacities (charge densities) and act quite differently with regard to FDR sensors. Unfortunately, only results from one clay type are shown and the expected comparison is never shown."

We agree with RC2's comment. Indeed, it is a pity to not be able to truly compare the results from FR-Lam and FR-Aur. The reason is that the samples from FR-Lam were not collected numerous enough and because, on FR-Lam, we do not use the "exhaust pipe clamp" and during the soil water content sensors probes insertion into the soil samples, we lost several soil samples. FR-Lam results are not complete and then, and only a qualitative behavior such as soil-specific calibration necessity or better homogeneity of the calibration factors on the surface, may be deduced. For these reasons, we do not compare results from FR-Lam with results from FR-Aur.

RC2: "The outcome that errors were larger for smaller water contents and smaller for larger water contents should be compared to results of others. All other studies of which I am aware show larger error at larger water contents

and relatively small error in dry soils. The authors found the opposite, and we need to understand why it happened. Was this a computational error?"

We are talking about *relative* errors, not *absolute* errors. When the absolute errors may decrease with SWC decrease, the corresponding relative errors may, on the contrary, increase. However, the papers we know dealing with the clayey soils with different factory calibrated FDR sensor checks, are not unanimous. Most of them show that the factory-calibrated sensor overestimates SWC but not all. Most of them show that the absolute error is decreasing with the SWC decrease but not all (see for example Lukanu and Savage: "Calibration of a frequency-domain reflectometer for determining soil-water content in a clay loam soil", DOI: 10.4314/wsa.v32i1.5237, where the factory calibrated FDR sensors show bigger absolute error at low SWC in clayey soil). Everything depends on what the factory-injected calibration factors are. The aim of this paper is not to discuss if the factory-calibrated sensor overestimates or underestimates the real SWC. This paper aims to show the necessity of the factory calibrated sensor check and how to calibrate FDR sensors in clayey soil in a laboratory, not in a field.

RC2: "The authors write that, "Once soil-specific calibration is done, FDR probes, and certainly other dielectric permittivity measurement-based probes, are accurate and may serve for SWC measurement." In general, this conclusion does not follow from the results given. The results given are for only one pit and one clay type. The authors should use the calibration equation for data from the other pits and show how well the calibration stands up when used for another pit and location with the same clay type. Then show how different the results are when the calibration is used for the other clay type (kaolinitic)."

There is a probable misunderstanding. We aim to show that the soil is specific not only for a plot but for any pit and any depth. Then "soil specific calibration" is a soil calibration for each pit and each depth. Each sensor should use a specific calibration factor depending on its localization. For example, we know that calibration factors calculated for pit A depth of 50cm will not necessarily fit the right calibrations factors for pit B on the same 50cm depth. The conclusion is rewritten to make this statement clearer. For accurate sensing, the best way is to make a soil-specific calibration for each planned sensor emplacement. It is impossible to provide a universal set of calibration factors. Several attempts are done to predict these factors from clay content, density, and so on. Up to now, to the best of our knowledge, the improvements of the SWC estimation by the *soil-predicted* calibrated sensors are not universal and the gain is limited. As stated in the conclusion: " Soil calibration is long and manpower-consuming but may be necessary."

RC2: "Lines 48-49: This is not true. The FDR sensors obey Gauss' law and thus are affected by capacitance. The TDR sensors obey Maxwell's equations and capacitance is not involved. Importantly, Gauss' law includes the complex permittivity, the bulk electrical conductivity, and a geometric factor that strongly affects capacitance. Maxwell's equations do not involve a geometric factor."

Indeed, the correct term would be "permittivity" not "capacitance". However, please, note that Gauss law does not include electric conductivity or geometric factors. Complex permittivity calculation comes from sensed complex impedance that implies complex bulk soil conductivity, capacitance, and then geometric factor. Maxwell's equation does not imply the geometric factors either, but the sensed signal does. We detail Gauss' law below in our answer to the RC2's comments (see lines 53-54).

RC2: "Lines 49-50: This is incorrectly written. The TDR method is a broad band method with central frequency in the 1 GHz range but the fast rise time pulse used in TDR methods is not emitted at frequencies of 1 GHz or even close to that."

We added a sentence about the wide frequency range.

RC2: "Lines 53-54: Again, incorrectly stated. An FDR sensor measures frequencies, which are affected by the capacitance of the soil-sensor system. According to Gauss' law, the capacitance is related to both the dielectric permittivity and the bulk electrical conductivity of the soil."

We are a little confused by the RC2 affirmation that the FDR sensors are "measuring frequencies" after pointing out that we have to make a clear difference between "sensing" and "measuring". FDR sensors measure *voltage* and deduct phase shift and signal attenuation between the emitted signal and the reflected sensed signal allowing the determination of a complex impedance. Also, lines 53-54 is stating: "An FDR sensor is measuring the soil capacity to store an electric charge directly related to the soil dielectric permittivity". Besides an eventual change of the word "measure" to "sense" and "capacity" to "capacitance", electric charge storage capacity implies some permittivity to create the charges with an electric field and low electric conductivity to not lose it by dielectric dispersion. It is not a Gauss law consequence but rather Ohm's law. Gauss' law written in the differential form is:

$$\nabla E = \rho/\varepsilon$$

With $\nabla$ nabla vectorial operator, $E$ electric field vector, $\rho$ charge density, and $\varepsilon$ absolute permittivity. There is no mention of conductivity or geometric factors. The passage between the sensed complex impedance and the dielectric permittivity does imply the geometric factor, not Gauss' law. Please note also that in Gauss' law the dielectric permittivity is not complex but a *purely real* number. Complex numbers introduction comes from AC signal analysis and fitting to a so-called "equivalent electrical circuit".
Anyway, our sentence does not contradict the fact that the electric conductivity causes the charge to vanish and affects the electric charge storage. Our sentence is only stating that the electric charge storage is directly related to the permittivity. This affirmation does not exclude other influences.

RC2: "Line 68: The clay soil for which the authors showed results could be termed ionic but it is important to understand that not all clay soils are ionic (highly charged). The kaolinitic clays have small charge and act more like sands with regard to their ionic and dielectric properties. Therefore, it is important for the authors to not state "For clayey soil...." but to make their statements specific to the soil with which they are working."

To specify, we added "FR-Aur" in the sentence.

RC2: "Lines 77-92: This is an odd way to begin a Materials and Methods section. This list of equipment could be given in a table, which would then be cited in the text explaining the method."

The section "M&M" was rewritten in the revised manuscript.

RC2: "Line 80: What is the meaning of "cloche" in the caption for figure 1 here? Would another word be more meaningful? A cloche is defined as a bell or dome-shaped cover. I do not see a cloche here."

As stated in the figure 1 legend the "cloche" is part "C" which is a "dome-shaped cover" held by a pneumatic percolator and covers the soil sampler. We kept the terminology "cloche".

[Figure]

RC2: "Line 117: What does "analogic probe tension" mean? Should it be "analog sensor voltage"? What are the units of voltage in the figure? Are these millivolts? Please give the units of voltage in the figure."

Effectively, "tension" is an old word replaced now by "voltage". The correct expression would be then "analog sensor voltage" and figure 3 b) was modified for volts.

RC2: "Line 119: Was there really only one reference digital TDR sensor used? Were there no replicates?"

Yes, only one reference sensor was used as the manufacturer asks to space out HydraProbes which would imply making several calibrations and each calibration is very slow (several months) due to the slow clayey soil evaporation. In this paper, we are assuming that the sensors used are identical to the sensors of the same models.

RC2: "Line 126: Which clayey soil was used for this? This is important. The results obtained with a kaolinitic clay would be different from those obtained with a smectitic or montmorillonitic clay. I see no reason to believe that the result (calibration) shown in Figure 3 is universally transferable among soils with different clay types and quantities. The calibration should be used with data from other soil pits to show if it is transferrable."

For all Fr-Aur soil samples, the digital and analog sensors were cross-calibrated using the FR-Aur soil.
The calibration of the analog sensor was necessary for the use of a sensor that could be disconnected for precise weighing without the sensor withdrawal from the sample. Especially in clayey soil, probes removal from the soil may be highly difficult, soil sample destructive or even sensor rods destructive where a back insertion is attempted (see for example Lukanu and Savage, reference added to the text). This could be avoided using sensors such as ML3 from Delta-T but, unfortunately, ML3 does not provide the real part of the permittivity. Another solution would be to insert a connector on the digital sensor cable but this would definitely prevent the modified sensor from being buried into the soil (real field use) and the relatively high cost of the digital sensor does not allow us to sacrifice several sensors. We are aware that the analog sensor calibration is not transferable between different soils. This sentence was explicitly added to the text (see lines 183-193). We checked the cross-calibration using the clayey soil from FR-Aur and sandy soil. The cross-calibration curves were obtained with different ranges of voltage for the analog sensor and of the real part of the permittivity for the digital sensor. However, the polynomial fits of those curves are close to each other, showing that this cross-calibration is not very sensitive to the soil texture.

RC2: "Lines 190=191: What does "with SWC (in $m^3/m^3$) increasing from 7% to 35%" mean? The SWC is in units of $m^3/m^3$. What does 7% to 35% mean?"

Formally, "$m^3/m^3$" is not a "unit" as [SWC($m^3/m^3$)]=1 (here "[]" is an operator) the SWC is calculated with a quotient of water volume (in $m^3$) divided by total soil volume (in $m^3$) is then "unitless" (the quotient is unitless) and could be calculated using any volume units such as $cm^3$, liters, gallons or $in^3$ without affecting the resulting SWC as long as both, water volume and total soil volume used in the quotient, are expressed with the same units. SWC is a "pure number" and unitless. Formally, specifying that the SWC is "volumetric" is enough and we do not need to add ($m^3/m^3$).
Percentage "%" is neither a unit and percentages are unitless. This notation is reserved then for unitless numbers and widely used for numbers varying between 0 and 1 reflecting a quotient or relative entities which is the case of volumetric SWC.
For example, the relative air humidity RH is usually given in % when RH=(water vapor partial pressure)/(saturation water vapor partial pressure) it means in (Pa/Pa) where "Pa" is a Pascal, pressure SI unit.
In other words, 1% = 0.01, so when we indicate volumetric SWC=7% it means that SWC=0.07 ($m^3/m^3$). We agree that this notation may be confusing so we have replaced "7%" with "0.07 ($m^3/m^3$)" and "35%" with "0.35 ($m^3/m^3$) and "%"  with m3/m3 when necessary".

RC2: "Lines 197-198: Please put the description of calculating the relative error and equation 3 at the beginning of the paragraph before Figure 6 is cited. Doing so will help the reader understand what is being discussed in the text."

The description of the relative error calculation is moved to the M&M paragraph (see line 228).

Comments which are included directly in the text.

RC2: line "This procedure is not very convincing. What precautions were made to prevent soil compression during sampling? Was the surface of the soil inside the tube compared with the surface of the soil in the pit wall outside the tube to determine if compression had occurred? What characteristics of the sampler design would have minimized compressive forces?"

An "optimized" soil sampler design description was added to the text as a sectional draw (see Fig. 2b). Please note that we are in clayey soil. As each pit is artificially dug, the surface of the pit wall is perfectly smooth, mirror-like, without any visible porosity. At a glance, there was no difference between the soil sample surface before and after extraction. The sampler has to be optimized to avoid sample compression during the sampler insertion and the sample extruder has to act slowly as the wall frictions between the soil and the sampler inner surface are viscous, increasing then with the differential velocity. The extruder piston surface should be very close to the soil sample surface to exert forces on the whole sample. As also answered to the RC1, when the sampler was enforced to the soil, the soil sample surface inside the sampler was at the same level as the surrounding soil surface which was plaid for compaction exemption.

RC2 line 187: "Should be real rather the "relative"?"

We noticed that in many publications the described "permittivity" is "unitless" and noted "ε". Formally the "permittivity"name is reserved for "absolute permittivity" which is not unitless and holds F/m (Faraday by meter) units. It can be expressed as a factor of "relative permittivity" usually noted as "$\varepsilon_r$" and vacuum absolute permittivity noted as "$\varepsilon_0$": $\varepsilon=\varepsilon_r\varepsilon_0$

This common abbreviation does not interfere with the understanding of the text. However, the correct name for a unitless permittivity indicated by most of the FDR is "relative susceptibility".
In the "theory" section a brief description of the used terminology was added (see section 2).

RC2 line 188: "What is meant by "modulus" here? Is this a part of the complex permittivity or is it the complex permittivity??"

As the estimated permittivity is a complex number, it means it is formed by a real part $Re(\varepsilon)$ and an imaginary part $Im(\varepsilon)$:
$\varepsilon= Re(\varepsilon)+i\ Im(\varepsilon)$ with "i" a *pure imaginary* of a specific property: $i^2=-1$. The "modulus" of $\varepsilon$, also noted $|\varepsilon|$, is:
$|\varepsilon| = \sqrt{Re(\varepsilon)^2 + Im(\varepsilon)^2}$. For more clarity, definition of the modulus was added into the section 2. Please note that a permittivity modulus is equal to the permittivity itself $|\varepsilon|=\varepsilon$ ($\varepsilon$ is a positive number) only in the case of a *pure real* permittivity ($Im(\varepsilon)=0$) which would be the case of the soil with a null conductivity.

---

## Referee Report (RR1)

**Responses to reviewers RC1 comments:**

We are sincerely grateful to our first referee hereafter called RC1 and to Steven Evett, hereafter called RC2, for taking their time to read, comment and help to improve our manuscript.

**General**

The authors reworked the MS with a lot of efforts and generally responded adequately on the remarks and suggestions. Language checking has been performed too, as was proposed by both reviewers.

Below are evaluations or remarks (in orange) on the responses (in blue).

RC1: "Introduction needs to be extended including more recent references and findings on this topic. At the end of the introduction readers expect clear objectives of the study, or a list of research questions. These need to be addressed in the Discussion and Conclusions."

References were added. The objectives have been clearly stated at the end of the introduction and addressed in the discussion and conclusion sections.

References were indeed added in the text, but no reference list was found at the end of manuscript version 2, so I could not check the full references…

Indeed, objectives were formulated (paragraph 75) and tackled in Discussion and Conclusion.

RC1: "In Material and Methods, clearly describe the test sites and sampling procedure, and inform the reader of the type of soil and its clay and organic content by depth layer in a table. Also, clearly describe the laboratory setup and refer to the protocol in Appendix A. Describe the statistical analysis, and introduce some measures for assessing bias and precision to evaluate the recalibration performance".

The M&M section was rewritten, reorganized and enriched with the requested information. We do not have soil analyses for each pit and depth. However, we included table 1 with the soil data that we have for FR-Aur site only. It is beyond the scope of this paper to establish a correspondence between soil composition, soil texture or soil density, and the calibration constants. The scope of this paper is only to suggest a check and soil-specific calibration using the soil of each planned SWC sensor implementation. Thereby, we do not need to know why the commercial sensors with the factory calibration constants overestimate or underestimate the real SWC. As it is rather difficult to work with clayey soil, we are suggesting a protocol that allows us to successfully conduct a calibration campaign. The introduction and conclusion were clarified. The quality levels of the calibrations were evaluated using the determination coefficient between soil-specific calibrated SWC estimation and Real SWC ($R^2$, see Eq. 4.). Then, the error made on SWC estimation relative to real SWC with the use of analog or digital FDR sensors factory settings was calculated using Eq. 3. There is not one emplacement where a factory-calibrated sensor is providing SWC within ICOS mandatory 0.05 m3/m3 accuracy. Different locations imply different calibration coefficients, which may be close to each other or not, unpredictable before the actual local calibration.

OK Thank you for rewriting this section. Still a bit strange that you do not have texture data for both sites, the more since this study is all about recalibrating the sensors due to the high clay content.

But you added references describing the general soil texture and clay mineralogy on the region, so this is already informative to the reader.

TYPO ! Near P150 "(at filed capacity)" => must be  field capacity

P165 Soil samples were collected in duplicate

Thanks for adding in Table 2 the SWC sensors specifications.

RC1: "In Results & Discussion section use Tables to evaluate the performance after recalibration, but also to show the difference between using the real part or modulus of dielectric permittivity

We added table 3 to support the performance of FDR sensors after recalibration. We also extended the results to a comparison between real part and modulus of the permittivity-based sensing.

Great !

RC1: "Provide in Discussion some answers to "What is acceptable accuracy of SWC measurements?" and "What is the minimum set of replicate samples per depth needed for proper recalibration?"

According to the ICOS protocol (Op de Beeck and al. reference added), the SWC probe should have at least 0.05 (m3/m3) accuracy over the whole expected SWC range sensed into four pits. We added this information inside the introduction and Mat & Meth sections (see lines 68 and 129).

Thank you for adding this accuracy threshold. It is very important to indicate that without soil specific calibration you cannot meet this level.

In paragraph 310 you write: "For example, for a real SWC value of 0.25 $m^3m^{-3}$ at 30 cm depth, the relative error decreases to -5.6 and -5.5 % for estimated SWC with digital and analog FDR sensors respectively, with $R^2$ values of 0.987."

So, then then the absolute bias is ~ 0.014 $m^3m^{-3.}$ Can you confirm that over the whole SWC range and depths, you are well below 0.05 $m^3m^{-3.}$  If so, I would put it explicitly in the text and conclusions, because then your goal is reached.

RC1: "In Conclusion: how are your research questions answered and objectives reached ?"

We have rewritten the conclusion to reflect more clearly how the objectives of our research were met.

RC1: "Below you can find comments by Line, also for the numerous typo's. Hope these can help you to improve the MS."

We are grateful to RC1 for his suggestions and we modified our manuscript in consequence.

RC1: "L26. Accuracy of 3%, do you mean 3% absolute (so volume %), or 3% relative compared to the gravimetrically determined SWC ?"

The value of "3%" is coming from the SWC sensors manufacturer's manuals and some of them, but not all, present that this is an absolute error of 0.03 m3/m3. We modified this value to the "0.03 m3/m3" value (see line 32)

RC1: "L27. What do you mean by "points" ? Measurement locations ?"

"points" means "other soil features" cited in the next sentence. Text modified (see line 34) RC1: "L37. Remove "usually", OM is never represented in a soil textural triangle" We are aware that OM is never represented in a soil textural triangle. We modified the sentence to clarify our message (see line 46 ).

RC1: "L71. choose the right probes for specific soils such as clayey soils => what are the specification of such rods ? Explained further ?

" This point was raised in the previous sentence "ionic soil, such as clayey soil, requires a real part dielectric constant based probe." This point was developed for better clarity. (see lines 109-113) Table 2 with sensor specifications was added. RC1: "L75. but even for a particular pit and particular depth for accurate SWC measurements. => see paper: https://www.mdpi.com/2076-3417/11/24/11620"

This reference was added into the text. RC1: "L79-85: I suggest to put the applied sensors in a table listing their characteristics (also rod lengths), as for example in Table 1 of https://essd.copernicus.org/articles/12/683/2020/"

Characteristics of sensors were added in table 2. RC1: "L86-92. Nice 'home made apparatus" – Have you checked bulk density differences compared to conventional volumetric sampling ? is the sample really taken by pneumatic hammering ? Or just pneumatic forcing into the soil ? What if there is a stone content in the soil ? How do you cope with samples containing coarse fragments ?"

Point by point:

- Thank you.

- The conventional volumetric sampling we know is to enforce a collar of a known volume into the soil, withdraw it, and crush the contained soil to liberate it from the collar making it unusable for further SWC measurement. Both methods use a collar or a sampler forced to the soil. The sampler forced into the soil may seem to be highly invasive and potentially compact the soil sample. However, the sampler was designed to minimize eventual compaction (a figure with the sampler sectional draw was added). Also, when the sampler was enforced to the soil, the soil sample surface inside the sampler was at an equal level as the surrounding soil surface which was plaid for compaction exemption.

OK

- Sampler enforced with a 5j perforator (added to the text).

- Stones are present on the Fr-Aur station (a few percent), We have a chance to not encounter any stone problem, nor during sample collection or further calibration process. However, it was planned that in case of sample collection impossibility due to a stone to collect another adjacent sample. In the case of SWC sensor insertion to the soil sample impossibility, it was planned to use the second

sample (two samples by location were withdrawn). Of course, it is not impossible that both samples contain stones preventing them to be used but this was highly improbable on FR-Aur. - The aim of the soil sample collection and the further use for calibration was to work as close as possible to the real conditions. Consequently, we do not discard any coarse elements from the soil samples. As we work on clayey soil, the only coarse elements are the stones.

OK, but since stone content was low, this was not really a problem.

RC1: "L95. So from Figure 2 I conclude you have 4 replicate SWC pits in FR_Aur and 5 in FR-Lam ? Please specify. Landuse is cropland ?"

Correct, we have four pits on FR-Aur and 5 pits on Fr-Lam and both stations are cropland stations. Specifications were added to the text.

Thanks !

RC1: "L 98. Refer to ICOS programme (https://www.icos-cp.eu/)" Reference added into the figure 1 (ex fig. 2) description.

RC1: "L105. As an illustration for this paper, some FR-Aur results are shown. Where ? Please refer to figure or table".

The whole text of this paper presents only the results of the FR-Aur soil calibration for more clarity. FR-Lam soil calibration is qualitatively comparable but not complete. RC1: "L106. Why are samples taken differently in the topsoil (vertically) compared to the subsoil layers ? I would take them all horizontally for a study. "

 Explanations were added to the text (see lines162). The "surface" sensors are vertically enforced to the soil and other depth sensors are placed horizontally so soil samples were collected according to the SWC sensors' placements.

RC1: "L105-109. Here you do not mention any hammering ? Just pressure."

Word "forced" was added to the text (see line 154).

RC1: " L110 "was near water-saturated" – water saturation, but probably you mean the soil was at "field capacity"."

The expression "at field capacity" was added to the text (see line 149). Please note that the "water-saturated soil" expression is also used.

RC1: "L119. List the properties of the analog and digital FDR probes in the instrument table please, so that the reader knows which devices these are."

See Table 2.  OK, thanks for adding.

RC1: "L128. Cracking is indeed one of the biggest problems of FDR measurements in clayey soils. You can avoid it in the lab, but how to cope with this in the field (especially topsoil) ?"

When the circularly distributed rod probes are used (it is the case with our sensors) the macro cracks mainly form around the probes as the soil is maintained by the rods. However, micro-crack formation is not avoidable. Not only the cracks forming between the rods but also around the rods with consequent poor electric contact between the rods and the soil. To our knowledge, there is no "magic" solution for that.

I am afraid you are right.

RC1: "L154 "Are the crack volume parts of the sample volume?" – typo "Crack" – by convention dry bulk density is the oven dry mass of soil (dried at 105°C) devided by its volume when taken in the field, mostly at field capacity. So in this sense, when a clayey sample dries out and is cracking, the crack volume is part of the sample volume, and no substraction is needed. Therefore it is called "bulk density", because it also includes pores, and channels, and cracks … in contrast to specific density of soil."

Absolutely. Then volumetric water content is the volume of the water divided by the volume of the soil including its cracks, any SWC sensing is then extremely localization depending. On a shrinking soil like our station's soil, cracks are often larger than the SWC sensor diameter. In the case of the presence of the crack any relatively small volume sensing device such as FDR, TDR, and so on provide a biased SWC estimation. Also multiplying the sensors does not help a lot since these sensors need to be inserted into the soil and do not work when one or more rods are in a crack. This is clearly a limitation for FDR and TDR sensors use in vertisol. Sentences were added to the text (see section 4.1).

I fully agree with this.

RC1: "L160. Figure 4. It would be informative to show progressive crack formation upon drying and this SWC- diel. Permittivity relationship."

Cracks formation observation is a challenge. There are three stages of the formation of the cracks: Vertical cracks formation visible on the surface, horizontal cracks formation inside the vertical cracks which are not visible from the surface, and vertical cracks formation inside the horizontal cracks which are not visible either. The link between the cracks and the permittivity would be a very interesting subject to study but it is definitely beyond the scope of our paper.

Indeed, maybe the topic for a next paper…

RC1: "This is probably linked with the fact that the tested soil samples originate from arable land that is homogenized by plowing. In forest and permanent grassland soils topsoil variability is usually greater and less homogenous."

Effectively, the apparent homogeneity of a surface soil is most probably linked with the soil tillage. The corresponding sentence was added (see lines 322).

RC1: " L170.(and L182) Figure 5 clearly shows overestimation of SWC by FDR, which has been reported by quite a lot of studies (e.g. https://doi.org/10.1016/j.agwat.2011.09.007)"

We agree that most of the studies showed an overestimation of the sensed SWC on clayey soil but not all. As the estimation error is determined by the factory-implemented calibration factors of the concerned sensors, everything is possible. Most of the sensors are factory calibrated for the most common soils and, with these settings, FDR sensors overestimate SWC assessments. However, the same sensors with different calibration constants may also underestimate SWC. It is just a question of settings.

RC1: "L195. How the relative error is calculated should be part of M&M section. Not in results."

Equation was moved to the M&M section (see line 228).

RC1: "L213. Once soil calibration is done, new calibration constants can be injected into the relations between SWC and the real part of dielectric permittivity. Clearly show how the recalibration is effectively performed, i.e; ; how soil-specific calibration coefficients are determined."

The specific calibration coefficients were determined by fitting the curves of "real soil water content versus indicated real part of the dielectric relative permittivity". Explanations were developed in M&M.

RC1: "L242. "calibration process is made during sample drying." For clayey soils, there is a hysteresis effect. Is calibration different when using the drying path compared to the rewetting trajectory ?"

The hysteresis is mainly due to the cracks' opening and closing. Cracks closing is favored by the internal soil pressure due to the surrounding soil. When a soil sample is used, there is no surrounding soil so the hysteresis observed with the soil samples is sensibly different from the hysteresis observed with real conditions. The rewetting was not attempted.

Yes, it is reasonable that under lab conditions real life hysteresis effects cannot be simulated, but it is likely that calibration will be different, no ?

RC1: "L258. Why are you not taking calibrated digital photo's to estimate the dimensions by digital image processing ? You are already taking photos for the cracks (L270)".

Sample diameter determination would be possible by digital photo processing but seems to us inaccurate especially when the sample soil is shrinking. The sample height would be not available as the samples are bucket surrounded.

RC1: "L38. "a soil-specific calibration may be required locally to determine the proper calibration of moisture versus dielectric permittivity constants" Is it better to do soil-specific calibration directly on the dielectric permittivity response of the sensors (as you did) or -in case of FDR, on derived sensor output signals like "period average" (or travel time in TDR) which also includes sensor characteristics ?"

As always, it is preferable to calibrate the whole process including as many characteristics as possible for precise calibration of a specific sensor. This calibration would be precise for only one model of a sensor. In case of any change in the sensor characteristic, all the calibration processes must be redone. The soil calibration allows to link dielectric permittivity with the SWC and does not allow to get rid of the sensor imperfection but is still valid for a "correctly" working sensor. "Correctly", means that the soil dielectric permittivity measurement by this sensor is accurate enough.

In this study, we are assuming that the main problem with SWC sensing comes from the relation between the permittivity and the SWC as this relation is soil-dependent. Indeed, factory injected

coefficients cannot be universal and soil-suggested calibration factors are not yet successful. However, a soil-specific calibration with the soil coming from the planned sensor emplacement, even if it is long, is still possible.

Thank you so much for your rebuttal. Well done !

---

## Author Response (AR2)

Dear Editor,

We agree with all the final remarks of the referee and added the eventual changes suggested. Concerning the final remarks about the hysteresis, if the hysteresis drying-rewetting is notable, the soil structure change, and the calibration process should be started again.

Best regards.